# The Penetration of Green Innovation on Firm Performance: Effects of Absorptive Capacity and Managerial Environmental Concern

**Min Xue [1], Francis Boadu [2,* ] and Yu Xie [2]**

[1] School of Management, China West Normal University, Nanchong 637009, China; xuem120016@cwnu.edu.cn
[2] School of Management and Economics, University of Electronic Science and Technology of China, Chengdu 611731, China; 201511110307@std.uestc.edu.cn
* Correspondence: linnacus@yahoo.com

**Abstract:** Under the background of environmental sustainability, it is of great significance to investigate how green innovation influences firm performance dimensions in emerging economies. Explicitly, the interaction effects of absorptive capacity (AC) and managerial environmental concern (MEC) on the correlation between green innovation and firm performance dimensions must be explored. Our data were obtained through a questionnaire survey from 253 companies operating in China. Using hierarchical linear modeling (HLM), we found that (1) green innovation has a robustly positive effect on firm performance dimensions (operational, financial and environmental), and (2) absorptive capacity and managerial environmental concern can positively affect the correlation between green innovation and firm performance dimensions. Our results illustrate the integrating effects of absorptive capacity, managerial environmental concern, green innovation and firm performance dimensions.

**Keywords:** green innovation; absorptive capability; managerial environmental concern; firm performance

## 1. Introduction

The 21st century has presented huge opportunities for both developed and emerging economies to benefit from development patterns. For, instance, economic growth by countries after World War II was made possible through dramatic upsurges in the use of material resources and energy. In the past few decades, countries in both developed and emerging economies have taken advantage of these development patterns. Emerging economies, specifically countries from Asia, have been experiencing high rates of economic growth through the introduction of material-intensive production and consumption patterns. However, in the past few years, it is apparent that these development patterns are causing calamitous environmental challenges (e.g., land degradation, environmental pollution, $CO_2$ emissions, etc.) towards sustainable economic development. Zhang [1] opines that the imbalance between economic growth and environmental sustainability is an albatross hanging around the neck of both developed and emerging economies.

Scholars and environmental strategists [2–4] have advocated and recognized that green innovation practices provide a critical tool for evolving sustainability practices. They believe that the adoption of a green innovation concept is a vital key in today's business activities to address environmental uncertainty. Stakeholders (e.g., consumers, suppliers, civil society organizations, etc.) have added their voices to pressure firms to adopt green innovation practices as a key driving mechanism to create a well-adjusted approach towards economic growth and environmental sustainability. This in turn can enhanced their corporate image [5]. Yim, et al. [6] refer to green innovation as a novel way of

doing things to accomplish or provide direct and positive benefits to the environment. Generally speaking, green innovation aimed at developing and implementing corrective measures to palliate environmental damages is needed [7]. Its role in addressing environmental challenges such as energy security, industrial pollution, climate change and the recycling of waste materials in today's hyper competitive environment has become progressively important for firms and societies. Admittedly, firms that infuse green innovation into their strategic agendas improve organizational and environmental efficiency [8–10].

From the extant literature, scholars have explored the direct influence of green innovation and its dimensions on firm performance. For example, prior research has examined the impact of green products [11,12], green processes [13] and green innovation [14,15] on firm performance. Unfortunately, some empirical analyses present inconsistent results among the variables [9,15–18]. These consequences still remain uncertain in the extant literature. Consequently, we argue that, lack of a cohesive logical basis or framework depicting the causal mechanisms through which green innovation spurs firm performance has not be established. Explicitly, the effect of absorptive capacity (AC) and managerial environmental concern (MEC) on green innovation and firm performance link must be determined.

Absorptive capacity, referred to as a firm's ability to recognize the value of new information, assimilate it and apply it to commercial ends [19], has been recognized as an important component of a firm's dynamic capabilities, as it enables firms to learn from partners, access external information and transform and integrate that information into existing knowledge stock. Previous studies have established a significant correlation between absorptive capacity and innovation consequences [19–21]. Conversely, managerial environmental concern reflects a firm's executives' commitment to environmental matters. Scholars like Przychoden et al. [22] and Qi et al. [23], postulate that managerial environmental concern can shape innovation activities in an organization towards enhanced performance.

We believe that the two variables (absorptive capacity and managerial environmental concern) can activate green innovation, which in turn enhances firm performance. However, the interplay among absorptive capacity, managerial environmental concern and green innovation towards firm performance is still not deeply explored. This intricacy underpins the need for practitioners and scholars to understand how these variables are interrelated. It is therefore prudent to examine whether or not, and if so how and when, an absorptive capacity and managerial environmental concern [24] facilitates (and moderates) green innovation and firm performance links. First, our study tests the direct correlation between green innovation and firm performance dimensions. Second, we suggest and discuss the effects of absorptive capacity and managerial environmental concern on the correlation between green innovation and firm performance dimensions.

Our study provides an extension to the literature. First, we contribute to the rising green innovation and firm performance literature by providing new insights into the mechanisms that influence operational, financial and environmental performances in organizations. Second, we respond to the call for more research to incorporate additional moderating variables—organizational slack and the environment—into the literature, to discover the connection between green innovation and firm performance [25,26]. Our findings have provided novel insight into the influencing mechanisms of absorptive capacity and managerial environmental concern on the link between green innovation and firm performance dimensions.

There are five parts in the study. Section 1 deals with the introduction. Theories and the development of hypotheses are described in Section 2. Section 3 tackles the empirical analysis adopted for the study. Section 4 deals with the empirical results and discussion, while the final section depicts the conclusion, limitations and future trajectory for further research.

## 2. Theory and Hypotheses Development

### 2.1. Green Innovation

Different authors have expressed their views on green innovation in different ways. Yim, Fung and Lau [6] describe green innovation as a transformation process that comprises novel ways of doing things (e.g., production–manufacturing, construction, procedures, systems, etc.) that provide direct and positive benefits to the environment. According to Chen, Lai and Wen [8], green innovation is a novelty used in technologies that incorporate energy saving, pollution prevention, waste recycling, green product designs and corporate environmental management. The import of the definitions centers around how stakeholders can adopt green technology to promote and attain organizational goals without hurting the environment. Admittedly, green innovation plays a major role in organizational development toward sustainability. Scholars [18,27,28] have acknowledged green innovation as a vital spark in a firm's performance management. In today's hyper competitive environment, firms aiming for survival should adopt effective green innovation policies and engage, build and develop to become relevant to stakeholders. With proper policies laid out, firms stand a chance to advance and become global players.

### 2.2. Absorptive Capability

The term absorptive capability was originally coined by Cohen and Levinthal [19]. They describe the concept as an ability to identify, integrate, transmute and apply external knowledge to commercial ends. These scholars assert that absorptive capability largely hinges on previous parallel knowledge or information, and multiplicity of background. For the past two decades, the work on absorptive capacity has developed significantly. It has been noted as a principal driving factor for firms implementing innovative plans [29,30]. The concept improves inter- or intra-firm learning [31], and also offers support to partners on the inter- or intra-firm network channel [32]. Firms' absorptive capacities can be considered as high or low. A high absorptive capability firm is assumed to be more proactive, innovative and ever-ready to assess and apply external knowledge resources critical in developing green innovation (product, process and organizational). Conversely, a low absorptive capability firm is presumed to be sedentary in hunting for external discoveries such as knowledge and technical activities, which are critical in developing green innovation.

### 2.3. Managerial Environmental Concern

The ever-growing concerns about the environmental consequences of business activities have raised a critical issue of environmental sustainability. In fact, the business executives or management can contribute to the reduction of potential environmental problems by resorting to good environmental management practices and guidelines in all business activities. Under this tendency, the decision to pursue environmental subjects by executives can be thought of as the choice between staying in environmental destruction versus environmental protection, based on the potential benefits available in either environmental destruction or environmental protection to the business. For firms to take this critical decision, a cognitive framing view offers management or executives a better understanding to opt for sustainability issues [33]. Admittedly, cognitive frames aid executives to filter all available information and select the best option [34]. Firms that are keen to promote green innovation opt for environmental protection or concern, and outline general management and environmental policies to promote organizational performance. Generally, firms can adopt two broad approaches (i.e., controlling and prevention) to manage environmental trepidations [35,36]. For example, the management team of an organization can curb or prevent an environmental menace by using all talents within a firm to develop themselves against environmental issues [37], in addition to the continuous development of existing production facilities and the introduction of new technological processes [9], as well as total quality management [37]. In this regard, firms' managerial environmental concerns can be considered as high or low. A managerial environmental concern firm is assumed to be more proactive

on environmental issues (e.g., environmental rules and regulations) and proffers innovative measures in developing green innovation activities. Conversely, a low managerial environmental firm is presumed to be passive or reactive to environmental matters (e.g., showing resistance to change). The next section sheds more light on the development of our hypotheses.

*2.4. Hypotheses Development*

We highlight the potential link among the identified variables in the study. Figure 1, below, illustrates the potential relationship among the independent variable (IV), the dependent variables (DVs) and the moderating variables.

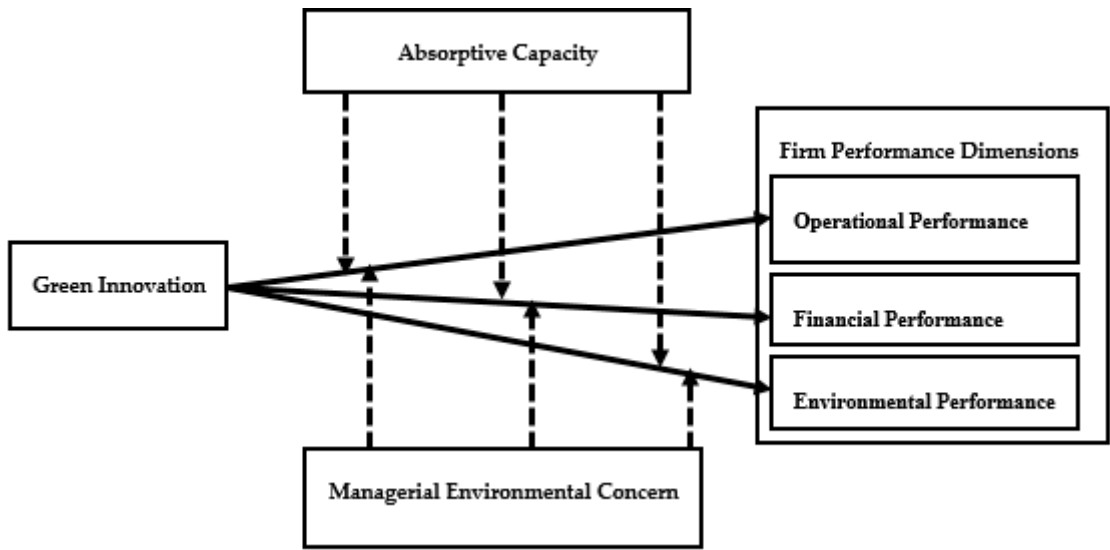

**Figure 1.** Logical framework.

*2.5. Green Innovation and Firm Performance*

Drawing on a resource-based view (RBV), a firm's resources are a crucial component for possessing a sustainable competitive advantage. For firms to uphold their sustainable competitive gains in today's dynamic market and surrounded by ever-swelling environmental pressures, espousing green innovation approaches can lead to offer valuable, inimitable and unique resources to outweigh rivals, especially in terms of performance. Nevertheless, the process cannot be achieved without contingent nature of knowledge generation, which is dominant for both business conceptions and market actions [38,39]. Erkut [39] posits that a new perspective christened "market shaping" plays a monumental role in enhancing innovation activities towards performance. He believes that innovation concepts adopted by firms lead them to carve market segments and benefits through technological and market knowledge, which may lead to success.

Firms that are keen on success always deploy arsenals in the market environment to assess how market actors are making use of business strategies. Indeed, the combination of knowledge gathered through this process and internal knowledge could be transformed into a pathway for the development of novel products and services to address market needs. Prior studies have been conducted by many researchers in both developed and emerging economies to establish the effects of green innovation on firm performance [17], but extant results among the variables present inconclusive reports [9,15–18]. Cheng, Yang and Sheu [9] posit that the application of green innovation in an organization has a positive effect to sustainable performance. Gluch, et al. [40] studied the Swedish construction industry, concluding that firms can adopt green innovations to enhance business performance. Scholars [8–10,41] assert that firms that incorporate green innovation into their strategic agendas improve organizational performance. Dangelico and Pontrandolfo [42] studied on the link between products and processes associated with environmental actions, and firm performances revealed a positive correlation effect.

However, studies conducted by other researchers produced adverse findings [9,15–18], leading their authors to the view that the link between green innovation and firm performance is indirect. Also, from theoretical and empirical perspectives, some scholars believe that green innovation is expensive, and for that matter, has a negative consequence on firm performance [43–45].

These discrepancies in the literature may be attributed to a lack of standardized measurement, methodology, difficulty in accessing data and underlying processes or contingency circumstances. In this study, we followed Chen, Lai and Wen [8] classification of green innovation (i.e., green product, green process and green organizational innovation), and used it as a bundle to examine green innovation's effect on firm performance dimensions.

### 2.5.1. Green Innovation and Operational Performance

Green innovation is a key driver for corporate and private organizations to meet their operational performance. It aids organizations to address critical environmental matters such as reduction of environmental impact, waste and $CO_2$. The concept also enhances productivity, corporate reputation and competitiveness [46–48]. A firm's ability to adopt green innovation practices to ameliorate pollution aids them to unfailingly achieve an aspect of operational purpose (e.g., cost reduction, purging of liability claims, etc.), which in turn leads to enhanced competitiveness. Imai, et al. [49] opine that the logic in averting pollution links to the quality management guiding principles that are critical to thwarting blunders at the premature phase, as opposed to amending after production. Therefore, we argue that integration of green talents, workforces within an organization and quality management plans with pollution prevention or protection strategies can go a long way to improving operational performance of a firm. Operational performance is referred to as a measurable aspect of the outcomes of an organization's processes (e.g., cost, production cycle time, inventory turns, reliability, flexibility, delivery dependency, quality and speed of new product introductions) [50].

Indeed, espousal of efficient and effective green integration practices is a critical strategy for achieving resource optimization, energy efficiency, environmental health and safety [51] and reliable production, which saves time and other related costs. These achievements present competitive advantage to firms to outweigh their competitors. Rusinko [52] postulates that green manufacturing operations are positively associated with a firm's competitive advantage. Scholars like Cainelli, et al. [53] and Horbach [54] have found a positive connection between green innovation and growth. Stakeholders assess a firm's green operational performance based on various components. These include competition (e.g., responsiveness to time of delivery), market dynamics (e.g., product quality development and line), cost, customer satisfaction and capacity utilization [55]. We believe that for green firms to remain relevant, they should sustain green operational performance to meet present needs without putting future generations' abilities on the line [56]. Pragmatic measures need to be in place to realize environmental sustainability. In view of the above discussions, we expect the following:

**Hypothesis 1a (H1a).** *Green innovation will be positively related to operational performance.*

### 2.5.2. Green Innovation and Financial Performance

Green innovation can make a positive impact on an organization outperforming its rivals. Firms mostly engage in green innovation activities to endorse more transactions that address the needs and wants of potential customers. This, in turn, can lead to better sales volumes [57] that enhance the financial position of the firm, although some empirical investigations indicate that the link among innovation and profitability is more complex to achieve [17]. Certain scholars [11,14,58,59] believe that green innovation has negative effects on the financial performance of a firm. In fact, they are of the view that firms that developing a habit of pursuing environmental objects can definitely end up with negative profitability [60]. We argue that a firm's green innovation enhancement efforts can upsurge the costs of their activities, which, in turn, erode profitability bases [58,59]. Aguilera-Caracuel and Ortiz-de-Mandojana [14] compared green innovation firms and non-green innovation firms on

financial performance, and discovered that green innovation firms do not enjoy enhanced financial performance. Liu, Dai and Cheng [44] opine that green innovation can lead to accrued costs. Driessen, Hillebrand, Kok and Verhallen [11] further added that green product innovation is allied with low financial performance.

Notwithstanding, most scholars [4,9,10,41,61] assert the positive impact of green innovation on financial performance. Fujii, et al. [62] investigated Japanese manufacturing firms, which provided a positive link between reduction of $CO_2$ emissions and financial performance. In addition, Bonini and Oppenheim [63] posit that eco-innovation practices contribute to a firm's financial performance. They believe that for firms to ameliorate waste and promote brands, the adoption of eco-innovation practices are essential for green-labeled branding that establishes status and will, in turn, attract potential customers and lead to financial gain. These inconclusive sentiments expressed by scholars need further investigation. For that reason, we expect to see that:

**Hypothesis 1b (H1b).** *Green innovation will be positively related to financial performance.*

2.5.3. Green Innovation and Environmental Performance

Various international environmental accords (e.g., the Paris accord on climate change, the Johannesburg declaration on sustainable development, the Montreal protocol on ozone layer depletion, etc.) continue to pressure countries to address the environmental menace caused by business activities on the environment. Green innovation policy is considered by scholars, stakeholders and policymakers as one of the crucial factors to ensure a win-win situation for businesses and the environment [64,65]. Performance-oriented firms apply green innovation concepts in their activities (e.g., processes, practices, systems) to provide direct and positive benefits to the environment. This plays a momentous part in the ecological sustainability spectrum.

Consequently, we argue that the implementation of effective and sustainable green innovation practices in an organization should attract all talents to tackle environmental deficiencies, which can enhance environmental benefits. In fact, such practices can reduce environmental menaces (e.g., air emissions, prevalence of environmental accidents) to enhance firms' environmental performances [66] and their images in the industry [42]. Indeed, the correlation between green innovation and environmental performance can be mediated by the high or low economic-financial results that firms obtain.

In the extant literature, some studies have reported conflicting views on the exact influence of green innovation on environmental performance. González-Blanco, Coca-Pérez and Guisado-González [60] conducted studies on Spanish manufacturing firms, demonstrating that green product innovation and green process innovation yields a dramatically negative influence on environmental performance. On the other hand, other authors have conducted investigations to establish a positive link between green innovation and environmental performance. For instance, empirical studies conducted by certain researchers [8–10,41,66] have attested that the infusion of green innovation into a firm's strategic agenda stimulates environmental benefits. Carrión-Flores and Innes [67], using panel data from the USA on 127 manufacturing firms, revealed a positive significant association between green innovation and environmental performance. In addition, Kim and Srivastava [68] found that green managerial innovation had a positive effect on environmental managerial performance. Thus, we state:

**Hypothesis 1c (H1c).** *Green innovation will be positively related to environmental performance.*

*2.6. The Moderating Effect of Absorptive Capacity and Managerial Environmental Concern*

This paper seeks to find an antidote to the paradoxical tendency that exists between green innovation and firm performance. Prior studies conducted by scholars [15,16] have established inconclusive reports on these two vital variables. Nonetheless, from the extant literature, some authors have long argued that a number of factors (e.g., organizational, contextual, etc.) might affect the

adoption of innovation in an organization [69]. For instance, for a firm to achieve efficacious innovation applications, organizational support plays an integral role in facilitating the process. Therefore, we argue that both absorptive capacity and managerial environmental concern can influence the correlation between green innovation and firm performance as predicted by H1a–c. First, we dwell on absorptive capacity, which is considered as an important driver for green innovation practices.

Scholars have rarely explored the moderating effect of absorptive capacity on the correlation between green innovation and firm performance. Absorptive capacity refers to a firm's ability to identify, integrate and hunt knowledge from its dynamic environment [19] to support organizational learning [70]. Thus, absorptive capacity aids firms in identifying and obtaining external resources through collaboration and partnership to augment internal resources for effective performance. Firms that are endowed with a sophisticated absorptive capacity have a tendency to advance a competitive advantage by means of innovation and a grander performance [71]. Admittedly, the notion of absorptive capacity affects variation in a firm's resource base. Firms' absorptive capacities can be considered as high or low. A high absorptive capability firm is assumed to be more proactive, innovative and ever-ready to access and apply the external knowledge resources that are critical in developing green innovation (product, process and organizational). Conversely, a low absorptive capability firm is presumed to be sedentary in hunting for external discoveries such as knowledge and technical activities, which are critical in developing green innovations.

The execution of green practices in an organization itself is the interface flanked by firms and the external environment. Indeed, it is a consolidation of both external and internal resources that leads firms to become accustomed to the turbulent environment and generate differentiation [9]. Stanovcic, et al. [72] emphasize that acquisition of external knowledge improves green innovation practices, and therefore firms should take critical steps to enhance their absorptive capacity [73]. Hashim, et al. [74] report that firms should stress the process through which knowledge influences green innovation. Huang and Li [75] discovered a positive significant association among absorptive capacity and green innovation development. We argue that the configuration of absorptive capacity and green innovation practices within an organization plays a vital role in enhancing performance. Thus, a firm's engagement of external knowledge into their organizational practices, especially in green innovation practices, can affect vicissitudes in the overall performance of the organization. We expect the following:

**Hypothesis 2a (H2a).** *The greater the absorptive capacity, the stronger the positive effect of green innovation on a firm's operational performance.*

**Hypothesis 2b (H2b).** *The greater the absorptive capacity, the stronger the positive effect of green innovation on a firm's financial performance.*

**Hypothesis 2c (H2c).** *The greater the absorptive capacity, the stronger the positive effect of green innovation on a firm's environmental performance.*

Second, in addition to absorptive capability, executives' understanding of environmental subjects is an essential issue for decision making in environmentally sensitive industries. Proper understanding unlocks avenues to filter all available information into, and allows executives to opt for the best choices for the betterment of the organization. Organizational inspiration from upper management shows influential effects on the readiness to approve green innovation practices [76]. Certainly, managerial environmental concern is one of the cardinal factors that can affect the espousal of green innovation in an organization [23]. Qi, Shen, Zeng and Jorge [23] studied the impact of managerial concern on espousing green innovation in the Chinese construction industry, and reveal that managerial concern is the most central driver for the espousal of green innovation. Przychodzen, Gómez-Bezares, Przychodzen and Larreina [22] assert that firms should consider the influencing factor of management in the conversion of green practices linked to firm performance. We believe that managerial environmental concern can activate green innovation, which, in turn, improves firm performance. In this regard, firms require

strong environmental policies that are embedded with norms, beliefs and values [77] to drive green innovation adoption towards performance and become competitive [78,79]. We state the following:

**Hypothesis 3a (H3a).** *The greater the managerial environmental concern, the stronger the positive effect of green innovation on a firm's operational performance.*

**Hypothesis 3b (H3b).** *The greater the managerial environmental concern, the stronger the positive effect of green innovation on a firm's financial performance.*

**Hypothesis 3c (H3c).** *The greater the managerial environmental concern, the stronger the positive effect of green innovation on a firm's environmental performance.*

## 3. Empirical Analysis

### 3.1. Research Setting

To test the hypotheses, we targeted heads of Chinese companies to assess how green innovation influences firm performance. Specifically, the interaction effects of absorptive capacity and managerial environmental concern on the correlation between green innovation and firm performance dimensions were explored. We decided on Chinese companies because of the following reasons: First, on the list of emerging economies, China is ranked as number one due to its rapid economic growth. In fact, the country is one of the biggest economies of our time, and an efficiency-driven one, according to the global competitiveness index [39]. It has grown to significant levels over the past few years. Second, China is ranked as the world's largest $CO_2$ emitter, due to its manufacturing activities. Third, over the past few years, the Chinese government has made frantic efforts towards environmental-sustainability by ratifying numerous international environmental accords (e.g., the Climate Change Agreement—COP 21 Paris, 2015) alongside internal environmental policies to address the environmental menace stemming from business activities. Indeed, authorities and consumer pressure groups are pushing and supporting the environmental go-green program to curb or reduce the environmental impact of business activities. Interestingly, this makes China an ideal place to investigate this phenomenon and to establish the exact effects of green innovation on firm performance dimensions.

### 3.2. Sample Size and Procedure

All Chinese companies targeted for this study were investigated from October to December 2018. Four hundred Chinese firms from the service, manufacturing and agriculture industries were selected as the scope of our research (see Appendix B). Using an intentional sampling technique, each head (i.e., director) was served with a questionnaire (see Appendix A) to respond to. We consulted the top executives due to their vast knowledge and expertise in their firms' overall operational activities. Executives reported their demographic details, information on green innovation and firm performance and integration of absorptive capacity and managerial environmental concerns. We adopted the ethical standards of the American Psychological Association and assured all executives about the confidentiality of the survey.

We retrieved 260 questionnaires in sealed envelopes from the firms contacted. This represented a 65% response rate for the analysis. All received surveys were examined to delete the incomplete ones with missing values. After the exercise, we retained 253 questions, representing a response rate of 63.25% for the final research (see Appendix B).

Prior to the survey, five experts in the area of innovation, environment and strategic management were engaged to review the content clarity, readability and comprehension of the questionnaire for the study. In addition, 40 firms in Western region of china that had similar characteristics to study subjects were used during the pre-testing period, to gauge the consistency of the questionnaire. All comments and suggestions expressed by the experts and the respondents were incorporated into the final assessment.

*3.3. Construct Measures*

The instrument used in this research was a survey questionnaire comprised of closed-ended questions (see Appendix A). It was segmented into five parts, covering 38 items. All items on the survey questionnaire were rated on a seven-point Likert scale (totally disagree = 1 to totally agree = 7), in order to measure the interrelationship among green innovation, managerial environmental concern, absorptive capacity and firm performance.

### 3.3.1. Explanatory Variable

We used green innovation as our independent variable. We adopted six items from Chen, Lai and Wen [8] to measure green innovation dimensions. Items included "Our organization frequently adopts new skills to develop novel green products to simplify their production and packaging", "Our organization frequently adopts new skills to develop novel green products to minimize damage from waste", "Our organization frequently adopts the latest production techniques to reduce waste", "Our organization frequently updates procedures to effectively reduce emissions of hazardous substances or waste", "Our organization frequently develops new tactics to update all relevant stakeholders on issues concerning green innovation during decision making" and "Our organization frequently develops new approaches to fit green innovation into management and administration to improve performance".

### 3.3.2. Dependent Variable

Firm performance is named as a dependent variable in this study. This contains a 10- item scale which reflects the entire performance dimensions (operational, financial and environmental) of the firm. The items were adopted from scholars [8,80,81]. Sample items comprise the following: "Compared to key competitors, our organization's responsiveness to time of delivery is better", "Compared to key competitors, our organization has improved product quality development and lines", "Compared to key competitors, our organization's return on sales have increased", "Compared to key competitors, our organization's ability to reduce air emission, waste and solid waste is better".

### 3.3.3. Moderating Variables

Our moderators were absorptive capacity and managerial environmental concern. First, absorptive capacity was used to measure the moderating effect on the link between green innovation and firm performance dimensions. We adapted four items from Omidvar, Edler and Malik [31]. Items included "Our organization has a capable or well-equipped staff to adapt newly acquired knowledge for development", "Our organization promotes an integrated approach to knowledge, sharing programs among different sectors", "Our organization has a capacity to understand and fuse assimilated external knowledge or resources for internal application", "Our organization has a capability to frequently change or revise quality control operations based on acquired new knowledge". Managerial environmental concern was used as our second moderating variable on the link between green innovation and firm performance dimensions. Four items were adapted from Ar [26]; Eiadat, Kelly, Roche and Eyadat [58]; Qi, Shen, Zeng and Jorge [23]. These items included "Environmental innovation is not necessary to achieve high profits", "Environmental innovation is an important component of strategy", "Most environmental innovations are worthwhile" and "Environmental innovation is an effective strategy".

### 3.3.4. Control Variables

To control our statistical results, the following variables were studied: firm size, firm age, industry type, ownership and financial subsidy.

First, prior works have found that firm size influences firm performance [82,83]. The size of a firm being small or large plays significant role in the assessment of performance. The larger ones tend to face greater pressure from government and other civil society organizations. These stakeholders

serve as a watchdog by discouraging nefarious activities that can create negative ecological effects. Conversely, because of their size, these firms can easily get access to resources (e.g., funding) to expand green innovation activities [84]. We controlled for firm size, indicated by the natural logarithm of the firm workforce (i.e., 0–199, 200–499, 500–999, 1000–4999 and >5000).

Second, prior studies [85,86] have found a connection among a firm's number of years in business and its performance. Mature firms have a tendency to perform higher than young firms. We controlled for firm age, indicated by the difference between year of establishment and date of questionnaire recovery.

Third, it is likely that the type of industry a firm operates in may affect performance, due to market and technological dynamics. For instance, market dynamics may force a firm to go for a particular green innovation activity to push its competitive plan in a specific industry. We controlled for characteristics of the industry that may be associated with firm performance (1 = manufacturing, and 0 = non-manufacturing).

Fourth, prior research found that ownership may have an influence on firm performance [87,88]. We controlled for ownership with state ownership coded as 1 and private ownership as 0 [89].

Finally, we controlled for financial subsidy, as prior studies [90,91] have established that it may be associated with firm performance. Firms were asked to indicate the financial subsidy that they obtain from authorities. This variable takes five values corresponding to the exact subsidy received by a firm (1 = 0–5M yuan, 2 = 5M–20M yuan, 3 = 20M–100M yuan, 4 = 100M–1B yuan and 5 = >1B yuan; M = million, B = billion).

### 3.4. Statistical Analysis

First, we conducted reliability, validity and confirmation factor analyses to determine the internal consistency, validity and the construct distinctiveness of the four variables (i.e., green innovation, absorptive capacity, managerial environmental concern and firm performance) of the study.

Second, we employed hierarchical linear modeling (HLM) to test the hypotheses of the study. This approach was necessary due to the multilevel, nested nature of the data [92]. According to Raudenbush and Bryk [92], HLM aids researchers in a variety of purposes (e.g., prediction, data reduction and causal inference from experiments and observational studies). The use of this technique in analysis allows for the existence of missing data at all levels except the highest level, the unequal measurement of time intervals, the ability to deal with hypotheses, and the assumptions of the homogeneity of variance and the independence of random error. However, HLM also has its drawbacks. Compared with traditional estimation techniques, this approach is more complex [93]. HLM is time consuming, as it requires more than three times the tracking data. The process allows for the accommodation of any number of hierarchical levels, and at times the workload of each level piles up, as most of the data have difficulty meeting necessary requirements.

For this paper, we thus followed prior HLM studies [92,94–96], and used the approach to analyze how green innovation can influence firm performance, as well as the moderating effect of absorptive capability and managerial environmental concern on green innovation and firm performance. Using SPSS 23.0, the direct H1a–c, and the moderating H2a–c and 3a–c (H2a–c; H3a–c), were tested. For H2a–c, we regressed the absorptive capacity on all control variables—green innovation, absorptive capacity and the interaction effect of green innovation and absorptive capacity (i.e., green innovation × absorptive capacity). Conversely, H3a–d focus on managerial environmental concerns. We regressed managerial environmental concern on all control variables—green innovation, managerial environmental concern and the interaction effect of green innovation and managerial environmental concern (i.e., green innovation × managerial environmental concern). Third, we followed Aiken, et al. [97] and plotted the interaction effects of green innovation on absorptive capacity and managerial environmental concern for the low and high levels of each moderator (i.e., the interaction at 1 SD below/above the mean).

## 4. Empirical Result

### 4.1. Test of Measurement

We conducted several tests in the areas of reliability, composite reliability (CR), average variance extracted (AVE) and Kaiser-Meyer-Olkin (KMO) to verify the suitability of all the four constructs (i.e., green innovation, managerial environmental concern, absorptive capacity and firm performance) of the study. SPSS23.0 was engaged to measure the constructs for fitness assessment. Table 1 presents the evaluation of the Cronbach's alpha, CR, AVE and KMO for each variable. The results of five Cronbach's alpha test, except operational performance ($\alpha > 0.6$) and CR, were all well above the threshold (0.70), suggesting a higher level of satisfactory and acceptable data according to Esbensen, et al. [98]. The value of AVE measuring the convergent validity indicated higher values, above 0.50 (50%). This demonstrates that the measurements correlated positively with the alternative measurements of the same constructs [99]. The values of KMO statistics ranged from 0.691 to 0.864, which was more than the accepted threshold of 0.6 [100].

**Table 1.** Construct reliability and validity.

| Constructs | Items | Cronbach's $\alpha$ | KMO | AVE | CR |
|---|---|---|---|---|---|
| Green innovation | 6 | 0.792 | 0.691 | 0.7119 | 0.8810 |
| Absorptive capacity | 6 | 0.843 | 0.864 | 0.5646 | 0.8856 |
| Managerial environmental concern | 4 | 0.745 | 0.756 | 0.5689 | 0.8399 |
| Operational performance | 4 | 0.697 | 0.712 | 0.5246 | 0.8148 |
| Financial performance | 3 | 0.790 | 0.707 | 0.7051 | 0.8776 |
| Environmental performance | 3 | 0.819 | 0.703 | 0.7353 | 0.8927 |

### 4.2. Descriptive Statistics and Correlation Test

A summary of the descriptive statistics data and correlation matrix is provided in Table 2. From that table, we can observe that the overall coefficient values were below 0.74, indicating that correlation coefficients were within the tolerable level of multicollinearity.

### 4.3. Hypotheses Testing

Hierarchical linear modeling was adopted to assess all hypotheses. H1a–c look at the direct link between the independent variable (IV) and dependent variables (DVs), while H2a–c and H3a–c look at the moderating role of absorptive capability and managerial environmental concern on the link between the IV and DVs.

#### 4.3.1. Main Effect Test

H1a proposes that green innovation will be positively related to operational performance. The coefficient for green innovation on operational performance is positive and significant ($\beta = 0.281$, $p < 0.001$, $R^2 = 0.246$), see Model 2 in Table 3. As such, H1a is strongly supported.

H1b predicts that green innovation will be positively related to financial performance. In our results from Table 4, Model 6 reveals that there is a positive correlation between green innovation and financial performance ($\beta = 0.306$, $p < 0.001$, $R^2 = 0.231$). Thus, H1b is verified.

**Table 2.** Descriptive statistics and correlations.

| Variables | 1 | 2 | 3 | 4 | 5 | 6 | 7 | 8 | 9 | 10 | 11 |
|---|---|---|---|---|---|---|---|---|---|---|---|
| Green innovation | 1 | | | | | | | | | | |
| Absorptive capacity | 0.034 | 1 | | | | | | | | | |
| Managerial environmental concern | −0.074 | 0.333 ** | 1 | | | | | | | | |
| Operational performance | 0.424 ** | 0.066 | −0.046 | 1 | | | | | | | |
| Financial performance | 0.405 ** | 0.026 | −0.055 | 0.598 ** | 1 | | | | | | |
| Environmental performance | 0.497 ** | −0.074 | −0.146 * | 0.224 ** | 0.235 ** | 1 | | | | | |
| Firm size | 0.311 ** | −0.073 | −0.201 ** | 0.307 ** | 0.287 ** | 0.308 ** | 1 | | | | |
| Firm age | −0.027 | −0.062 | −0.071 | −0.162 ** | −0.111 | −0.019 | −0.050 | 1 | | | |
| Manufacturing | 0.145 * | −0.115 | −0.132 * | 0.188 ** | 0.209 ** | 0.159 * | 0.199 ** | 0.179 ** | 1 | | |
| State ownership | −0.403 ** | −0.035 | −0.040 | −0.267 ** | −0.156 * | −0.242 ** | −0.104 | −0.310 ** | 0.114 | 1 | |
| Financial subsidy | 0.300 ** | −0.023 | −0.141 * | 0.216 ** | 0.277 ** | 0.325 ** | 0.103 | 0.093 | 0.227 ** | −0.060 | 1 |
| Mean | 4.882 | 4.723 | 4.639 | 4.690 | 4.564 | 4.661 | 2.581 | 3.000 | 0.435 | 0.391 | 3.281 |
| SD | 0.861 | 0.971 | 1.015 | 0.868 | 1.069 | 1.001 | 1.371 | 1.205 | 0.497 | 0.489 | 0.920 |

Note: * $p$ value < 0.05, ** $p$ value < 0.01; (two-tailed, $N$ = 253).

**Table 3.** Results of hierarchical linear modeling analysis (operational performance).

| Variables | Dependent Variable: Operational Performance | | | |
|---|---|---|---|---|
| | Model 1 | Model 2 | Model 3 | Model 4 |
| Firm Size | 0.229 (0.043) *** | 0.164 (0.043) ** | 0.164 (0.042) ** | 0.164 (0.042) ** |
| Firm Age | −0.128 (0.05) * | −0.146 (0.049) * | −0.124 (0.048) * | −0.123 (0.048) * |
| Manufacturing | 0.152 (0.121) * | 0.13 (0.118) * | 0.137 (0.116) * | 0.125 (0.116) * |
| State ownership | −0.211 (0.123) *** | −0.101 (0.13) | −0.076 (0.129) | −0.095 (0.128) |
| Financial subsidy | 0.159 (0.064) ** | 0.094 (0.064) | 0.056 (0.064) | 0.07 (0.063) |
| Green innovation | | 0.281 (0.066) *** | 0.278 (0.065) *** | 0.275 (0.064) *** |
| Absorptive capacity (AC) | | | 0.087 (0.054) | |
| Managerial environmental concern (MEC) | | | | 0.06 (0.056) |
| Green innovation × AC | | | 0.176 (0.059) ** | |
| Green innovation × MEC | | | | 0.191 (0.053) *** |
| R$^2$ | 0.194 | 0.246 | 0.275 | 0.277 |
| F | 13.125 *** | 14.739 *** | 12.933 *** | 13.049 *** |

*** $p < 0.001$, ** $p < 0.01$, * $p < 0.05$, (Standard error), $N = 253$.

**Table 4.** Results of hierarchical linear modeling analysis (financial performance).

| Variables | Dependent Variable: Financial Performance | | | |
|---|---|---|---|---|
| | Model 5 | Model 6 | Model 7 | Model 8 |
| Firm Size | 0.218 (0.043) *** | 0.147 (0.043) * | 0.144 (0.042) * | 0.148 (0.043) * |
| Firm Age | −0.116 (0.051) | −0.136 (0.049) * | −0.114 (0.049) | −0.114 (0.049) |
| Manufacturing | 0.145 (0.123) * | 0.121 (0.119) * | 0.124 (0.118) * | 0.117 (0.117) * |
| State ownership | −0.1 (0.125) | 0.02 (0.132) | 0.047 (0.13) | 0.026 (0.13) |
| Financial subsidy | 0.228 (0.065) *** | 0.157 (0.064) ** | 0.116 (0.065) | 0.134 (0.064) * |
| Green innovation | | 0.306 (0.066) *** | 0.307 (0.065) *** | 0.3 (0.065) *** |
| Absorptive capacity (AC) | | | 0.049 (0.055) | |
| Managerial environmental concern (MEC) | | | | 0.001 (0.057) |
| Green innovation × AC | | | 0.187 (0.06) ** | |
| Green innovation × MEC | | | | 0.177 (0.054) ** |
| R$^2$ | 0.168 | 0.231 | 0.258 | 0.256 |
| F | 11.175 *** | 13.599 *** | 11.964 *** | 11.825 *** |

*** $p < 0.001$, ** $p < 0.01$, * $p < 0.05$, (Standard error), $N = 253$.

H1c predicts that green innovation will be positively related to environmental performance. Model 10 in Table 5 shows that green innovation positively affects environmental performance (β = 0.349, *p* < 0.001, R$^2$ = 0.293).

**Table 5.** Results of hierarchical linear modeling analysis (environmental performance).

| Variables | Dependent Variable: Environmental Performance | | | |
|---|---|---|---|---|
| | Model 9 | Model 10 | Model 11 | Model 12 |
| Firm Size | 0.245 (0.042) *** | 0.164 (0.041) ** | 0.154 (0.041) ** | 0.15 (0.041) ** |
| Firm Age | 0.023 (0.05) | 0 (0.047) | 0.01 (0.047) | 0.014 (0.047) |
| Manufacturing | 0.069 (0.12) | 0.041 (0.114) | 0.034 (0.114) | 0.033 (0.113) |
| State ownership | −0.215 (0.122) *** | −0.078 (0.126) | −0.059 (0.126) | −0.077 (0.125) |
| Financial subsidy | 0.27 (0.063) *** | 0.189 (0.062) ** | 0.162 (0.063) ** | 0.163 (0.062) ** |
| Green innovation | | 0.349 (0.064) *** | 0.359 (0.063) *** | 0.344 (0.063) *** |
| Absorptive capacity (AC) | | | −0.062 (0.053) | |
| Managerial environmental concern (MEC) | | | | −0.079 (0.055) |
| Green innovation × AC | | | 0.118 (0.058) * | |
| Green innovation × MEC | | | | 0.140 (0.052) ** |
| R$^2$ | 0.210 | 0.293 | 0.306 | 0.310 |
| F | 14.415*** | 18.441*** | 14.868*** | 15.163*** |

*** $p < 0.001$, ** $p < 0.01$, * $p < 0.05$, (Standard error), $N = 253$.

### 4.3.2. Moderating Effect Test

First, H2a predicts that the greater the absorptive capacity, the stronger the positive effect of green innovation on operational performance. The results in Table 3, Model 3 show that absorptive capacity positively moderates the association between green innovation and operational performance ($\beta = 0.176$, $p < 0.01$, $R^2 = 0.275$). Thus, H2a is validated. Figure 2 demonstrates that the effect of green innovation on operational performance is stronger when absorptive capacity is high (e.g., 1 SD above the mean). In this case, the high absorptive capacity will push the simple slope corresponding with green innovation and operational performance to become stronger or larger. The effect is lessened when absorptive capacity is low (e.g., 1 SD below the mean).

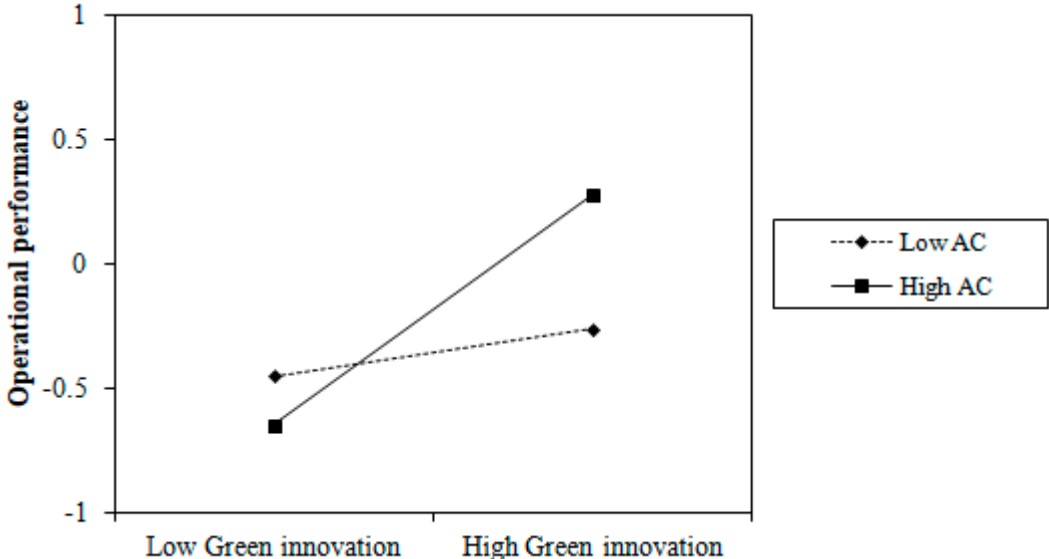

**Figure 2.** Absorptive capacity's moderating effects on green innovation and operational performance.

Second, in support of H2b, Model 7 in Table 4 shows that the interaction effect between green innovation and absorptive capacity has a positive significance ($\beta = 0.187$, $p < 0.01$, $R^2 = 0.258$). As shown in Figure 3, the effect of green innovation on financial performance is stronger when absorptive capacity is high (e.g., 1 SD above the mean). In this case, the high absorptive capacity will push the simple slope corresponding with green innovation and financial performance to become stronger or larger. The effect is lessened when absorptive capacity is low (e.g., 1 SD below the mean). Therefore, H2b is verified.

Third, H2c proposes that the greater the absorptive capacity, the stronger the positive effect of green innovation on environmental performance. The beta coefficient for the interaction effect between green innovation and absorptive capacity is significant and positive ($\beta = 0.140$, $p < 0.01$, $R^2 = 0.310$), see Model 11 in Table 5. Therefore, H2c is supported. In Figure 4, the effect of green innovation on environmental performance is stronger when absorptive capacity is high (e.g., 1 SD above the mean). In this case, the high absorptive capacity will push the simple slope corresponding with green innovation and operational performance to become stronger or larger. The effect is lessened when the absorptive capacity is low (e.g., 1 SD below the mean).

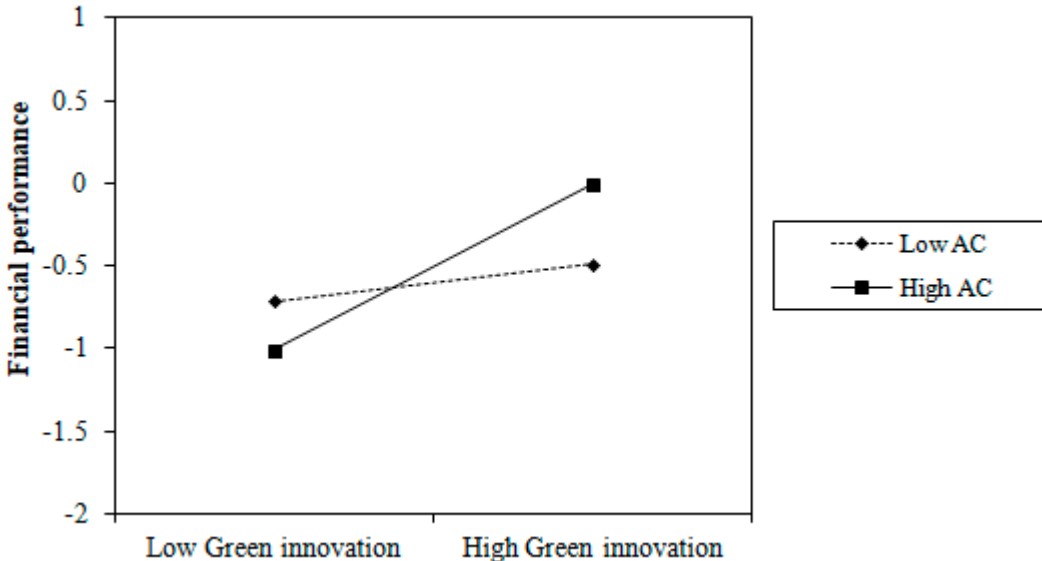

**Figure 3.** Absorptive capacity's moderating effects on green innovation and Financial Performance.

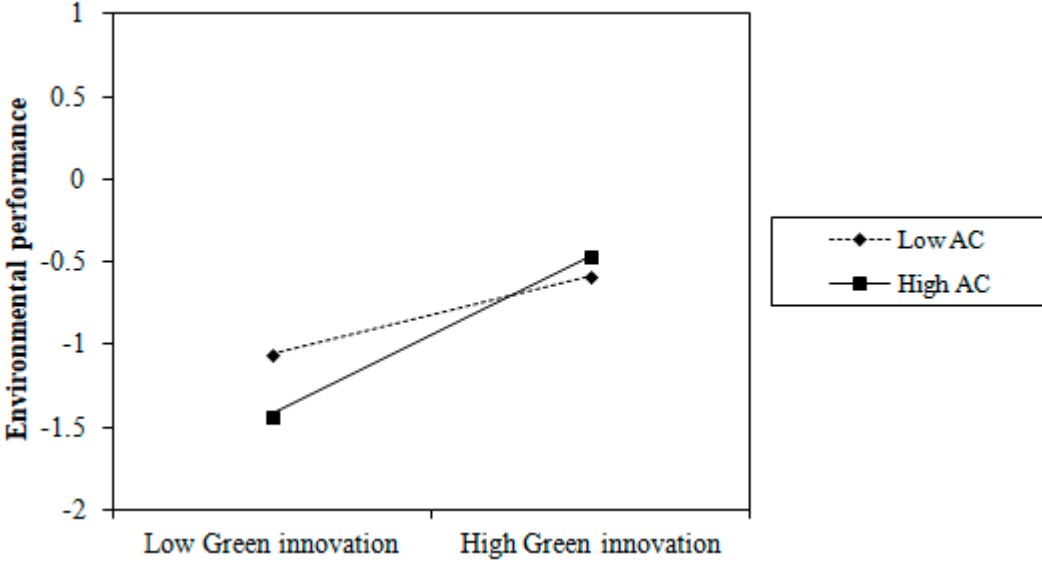

**Figure 4.** Absorptive capacity's moderating effects on green innovation and environmental performance.

Fourth, H3a predicts that the greater the managerial environmental concern, the stronger the positive effect of green innovation on operational performance. The results in Table 3, Model 4 demonstrate that managerial environmental concern positively moderates the association among green innovation and operational performance ($\beta = 0.191$, $p < 0.001$, $R^2 = 0.277$), supporting H3a. Figure 5 illustrates that the effect of green innovation on operational performance is stronger when managerial environmental concern is high (e.g., 1 SD above the mean). In this case, the high absorptive capacity will push the simple slope corresponding with green innovation and operational performance to become stronger or larger. The effect is lessened when managerial environmental concern is low (e.g., 1 SD below the mean).

Furthermore, H3b projects that the greater the managerial environmental concern, the stronger the positive effect of green innovation on financial performance. As shown in Table 4, Model 8, the F-statistic is significant (F = 11.825, $p < 0.001$, $R^2 = 0.258$), thus the moderation effect is significant. Figure 6 proves that the effect of green innovation on financial performance is stronger when managerial environmental concern is high (e.g., 1 SD above the mean). In this case, the high absorptive capacity

will push the simple slope corresponding with green innovation and financial performance to become stronger or larger. The effect is lessened when managerial environmental concern is low (e.g., 1 SD below the mean).

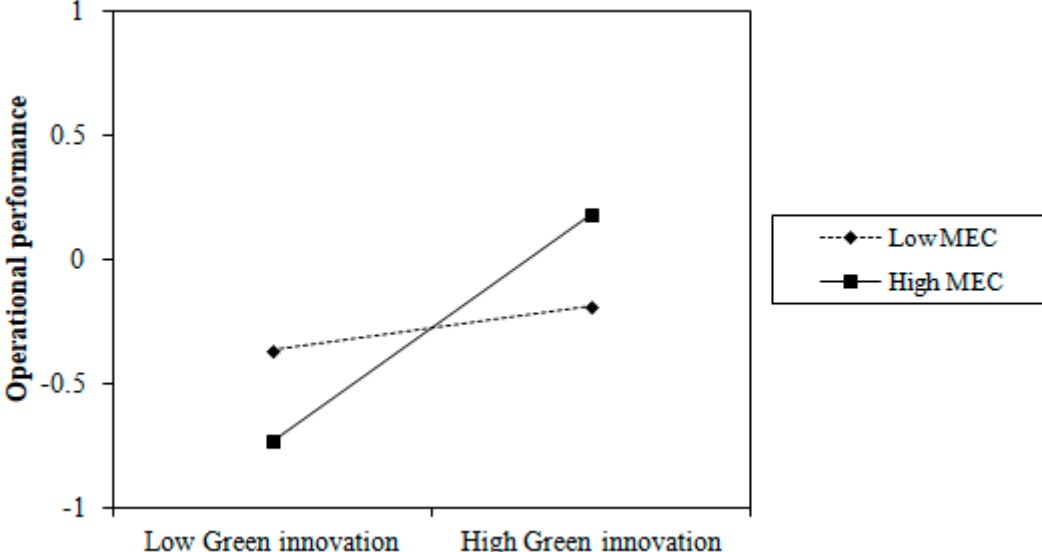

**Figure 5.** Managerial environmental concern's moderating effects on green innovation and operational performance.

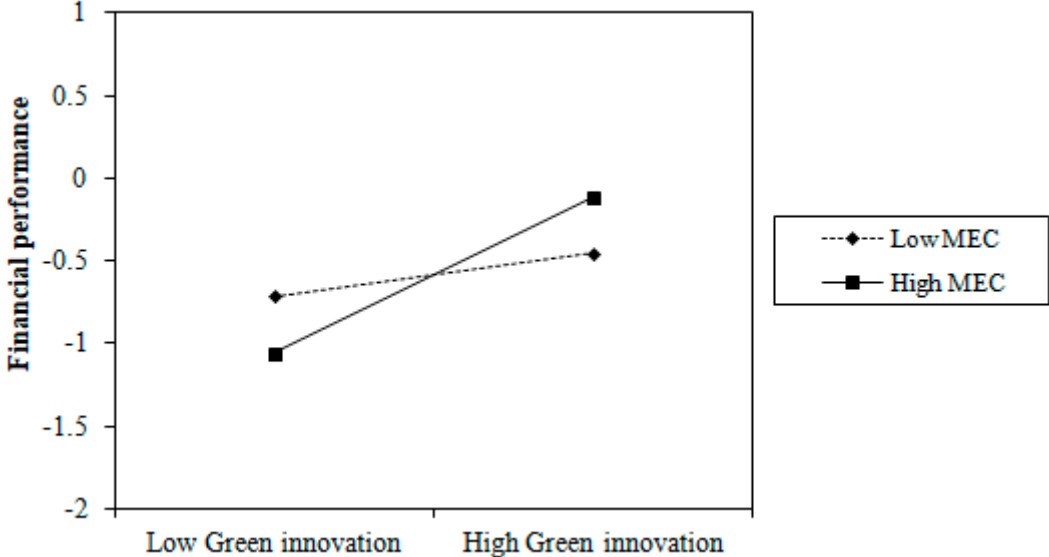

**Figure 6.** Managerial environmental concern's moderating effects on green innovation and financial performance.

Finally, in support of H3c, Model 12 in Table 5 demonstrates that the interaction effect among green innovation and environmental performance is significant ($\beta = 0.140$, $p < 0.01$, $R^2 = 0.310$). Figure 7 establishes that the effect of green innovation on environmental performance is stronger when managerial environmental concern is high (e.g., 1 SD above the mean). In this case, the high absorptive capacity will push the simple slope corresponding with green innovation and environmental performance to become stronger or larger. The effect is lessened when managerial environmental concern is low (e.g., 1 SD below the mean).

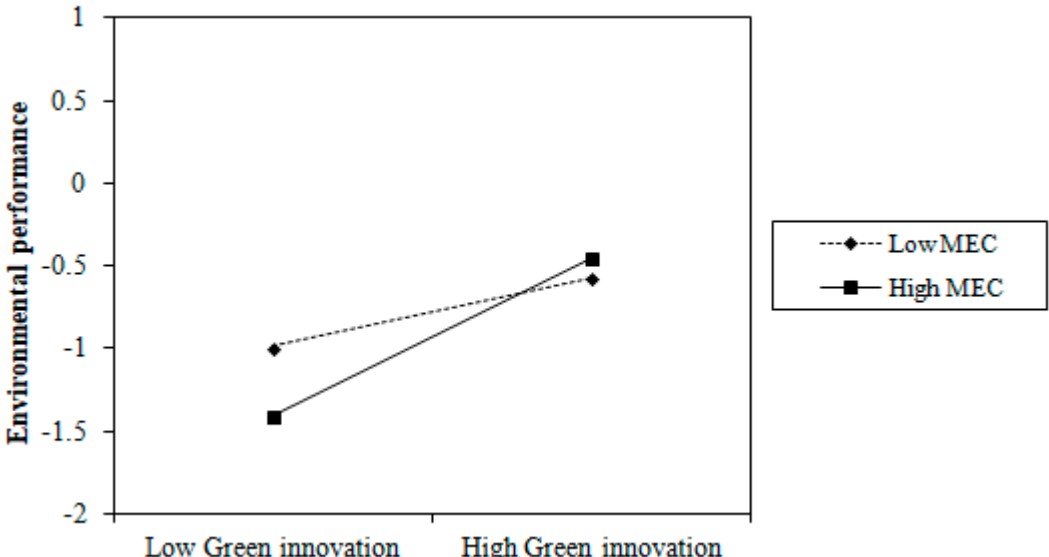

**Figure 7.** Managerial environmental concern's moderating effects on green innovation and environmental performance.

*4.4. Discussion*

This study advances existing theoretical and extant literature on two fronts. First, we offer a more refined scrutiny of the link between green innovation and firm performance. Most prior studies tend to treat firm performance as a unidimensional construct. In this study, we conceptualize firm performance as a multi-dimensional construct (i.e., operational, financial and environmental performances). Our results reveal a significant, robust effect among green innovation and these multi-dimensional constructs. Thus, we have provided new insights to lay foundations into mechanisms that might influence operational, financial and environmental performances in an organization.

Second, we respond to call for more research to incorporate additional moderating variables, specifically organizational slack and the environment, to discover the connection between green innovation and firm performance [25,26]. From the extant literature, scholars [69] have proposed that a number of factors (e.g., organizational, environmental, contextual, etc.) can influence the espousal of innovation towards performance. Our results confirm a positive influence of absorptive capacity on the linkage between green innovation and firm performance dimensions. Indeed, a firm's ability to source and apply external information (absorptive capability) plays a critical role in business growth and sustainability [29]. As fresh knowledge and ideas are generated from external sources through research and development (R&D) activities [29,101], firms can augment internal knowledge and technology stock to ameliorate the costs associated with the process of applying and sustaining green innovation in an organization. Therefore, firms with strong absorptive capacity stand a chance to benefit significantly from green innovation activities. These findings are in line with prior studies [71,102].

Equally important, the results show that the effect of managerial environmental concern on the link between green innovation and firm performance dimensions is significant. This indicates that managerial concern over environmental regulations and policies plays a significant role in firms' green innovation applications towards sustainable development. The willingness of management to observe and evaluate strategic relevance of green [18] provides greater managerial control in addressing environmental pressures from the government, customers and civil society groups towards implementation of green innovation. These findings are in line with prior research [23] that has established a key relationship among managerial environmental concern and green innovation practices. In the current enquiry, we have taken a step further to integrate the research streams of green innovation and firm performance by incorporating key moderating factors, which translate green innovation practices into performance.

This paper presents four valuable proposals for practitioners and policymakers. Our results can significantly impact structural similarities in China in terms of innovation with other efficiency-driven economies [39]. First, firms need to demonstrate organizational commitment and support for the improvement of environmental issues. Management should prioritize environmental activities and continuously follow and obey all environmental regulations and policies to create enviable sustainable development. In fact, prioritizing environmental activities is not simply a cost burden, but a feasible approach that can lead to competitive advantages in the market arena. Consequently, practitioners should invest in environmental programs to enhance the capability of workforces. This will, in turn, improve green innovation activities in an organization, leading to enhanced performance.

Second, firms' executives need to focus on absorptive capabilities for enhancing organizational activities. Practitioners must attempt to incorporate external knowledge that can complement the internal knowledge stock and address pertinent issues regarding green innovation activities. Thus, executives should create a platform that will allow the workforces from various units and other professional bodies to interact and share knowledge, ideas and other relevant issues to meet the demands of stakeholders (e.g., customers, government regulation authorities, environmental promoters, etc.). A firm's endorsement and application of this practice can boost their green innovation agenda, and allow them to become proactive in terms of providing innovative services and products that address consumer and environmental needs.

Third, policymakers should continue to create awareness of entrepreneurial activities concerning the environment. They should implement strict environmental supervision of firms' activities, which could stimulate green innovation. Indeed, such policy prescriptions could go a long way in creating a good environmental atmosphere towards sustainable development.

Fourth, policymakers should create an enabling atmosphere to support corporate green innovation. Effective policies and progressive measures (e.g., grants, tax holidays and rebates) can aid firms to curb environmental menaces such as pollution and emissions, among others.

## 5. Conclusions

Green environmental initiatives provide a monumental opportunity for a firm's growth, including enhanced performance and sustainability. Most firms in advanced economies have benefited from green activities, and position themselves in the international arena to the detriment of their counterparts from emerging and developing economies. Drawing from a quantitative study of 253 companies operating in China, this study investigated the direct correlation between green innovation and firm performance dimensions. Specifically, it investigated the interaction effects of absorptive capacity and managerial environmental concern on the correlation between green innovation and firm performance dimensions. Our study demonstrates that green innovation has a positive significant influence on firm performance dimensions. We also reveal that these effects are facilitated by absorptive capability and managerial environmental concern. These findings illustrate the integrating effects of absorptive capacity, managerial environmental concern, green innovation and firm performance dimensions. They supplement the theoretical and empirical exploration concerning the green innovation and firm performance of companies.

The present study provides an insightful contribution to the body of literature. However, it cannot move forward without overcoming certain limitations. First, the study cannot be generalized to cover entire emerging economies. Our findings stem from a Chinese context only, and therefore, the positive impacts of green innovation on firm performance dimensions, as well as the positive moderating effect of absorptive capacity and managerial environmental concern on green innovation and firm performance dimensions, may be more specific to this region. We encourage future empirical research to examine the above variables in other economies (i.e., both emerging and developed economies). A second limitation of the study is the cross-sectional research design adopted to establish the causal directions among the variables. We encourage future researchers to examine the variables through longitudinal or experimental studies designed to detect causal relationships in more details. Third,

this study uses subjective assessment to measure the financial performances of companies in China. It would be desirable for future research to use other measures (e.g., objective data) of financial performance. Such a study would allow us to verify whether the results obtained in our study are maintained. Fourth, other moderating variables from the environment may exist for the green innovation and firm performance relationship. We encourage future researchers to identify more systematic development to better specify the influencing mechanisms on the green innovation and firm performance link. Fifth, using the HLM methodology helps us overcome certain limitations in traditional statistical techniques [103] through increased access of existing missing data at all levels and assessment of multi-level relationships. However, the approach is complicated and time-consuming as compared with traditional statistical techniques. In addition, it presumes that data is normally distributed, but when the assumption of normality of the predictor and/or outcome variable(s) is violated, this range restriction biases HLM output. We encourage future researchers to explore multiple methods to investigate the interplay between the variables. Finally, while our study focused on how Chinese firms applied the interplay among absorptive capability, managerial environmental concern and green innovation towards firm performance, we encourage future researchers to replicate the variables, methodology and research design in other efficiency-driven economies (e.g., Brazil, Russia, South Africa, etc.) to confirm and enhanced generalizability.

**Author Contributions:** Conceptualization, M.X. and F.B.; Data curation, M.X., F.B. and Y.X.; Formal analysis, M.X., F.B. and Y.X.; Funding acquisition, M.X.; Investigation, M.X. and F.B.; Methodology, Y.X.; Writing—original draft, M.X. and F.B.; Writing—review & editing, M.X., F.B. and Y.X.

**Funding:** This study was supported by the Science and Technology Department of Sichuan Province (Grant No. 2019JDR0027) and the Special Project of Ph.D. Science and Technology by China West Normal University (Grant No. 18Q010).

**Acknowledgments:** We are grateful to all the funding agencies, the editors and the anonymous reviewers for valuable comments and suggestions. We also appreciate the effort of the business captains who agreed to participate in the survey.

**Conflicts of Interest:** The authors declare no conflict of interest.

## Appendix A  Questionnaire

1. **Green Innovation** Our organization frequently adopts new skills to develop novel green products to simplify their production and packaging. Our organization frequently adopts new skills to develop novel green products to minimize damage from waste. Our organization frequently adopts the latest production techniques to reduce waste. Our organization frequently updates procedures to effectively reduce emissions of hazardous substances or waste. Our organization frequently develops new tactics to update all relevant stakeholders on issues concerning green innovation during decision making. Our organization frequently develops new approaches to fit green innovation into management and administration to improve performance.

2. **Absorptive Capacity** Our organization has a capability or well-equipped staff to adapt acquired new knowledge for development. Our organization promotes an integrated approach to knowledge, sharing programs among different sectors. Our organization has a capacity to understand and fuse assimilated external knowledge or resources for internal application. Our organization has a capability to frequently change or revise quality control operations based on acquired new knowledge. Our organization invests a lot in research and development. Our organization frequently encourages staff to participate in regular training, exploration, discussion of market trends, new product development and scientific events (e.g., from conference, seminars, courses, etc.) for the discovery of useful resources.

3. **Managerial Environmental Concern** Environmental innovation is not necessary to achieve high profits. Environmental innovation is an important component of strategy. Most environmental innovations are worthwhile. Environmental innovation is an effective strategy.

4. **Operational Performance** Compared to key competitors, our organization's responsiveness to time of delivery is better. Compared to key competitors, our organization has improved product quality development and lines. Compared to key competitors, our organization has improved capacity utilization. Compared to key competitors, our organization's customer satisfaction is better.

5. **Financial Performance** Compared to key competitors, our organization's dividends to shareholders have increased significantly. Compared to key competitors, our organization's returns on sales have increased. Compared to key competitors, our organization's profits have increased over the years.

6. **Environmental Performance** Compared to key competitors, our organization's ability to reduce air emission, waste and solid waste is better. Compared to key competitors, our organization's reduction of environmental accidents is better. Compared to key competitors, our organization's environmental situation has improved.

**Appendix B  Characteristics of the Research Samples**

| Industry | Location | Number | Percentage |
|---|---|---|---|
| Service | Eastern Area of China | 62 | 24.51% |
|  | Central Region of China | 18 | 7.11% |
|  | Western Region of China | 47 | 18.58% |
| Manufacturing | Eastern Area of China | 68 | 26.88% |
|  | Central Region of China | 16 | 6.32% |
|  | Western Region of China | 26 | 10.28% |
| Agriculture | Eastern Area of China | 2 | 0.79% |
|  | Central Region of China | 8 | 3.16% |
|  | Western Region of China | 6 | 2.37% |
| Total |  | 253 | 100% |

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
