# Peer review of "The Penetration of Green Innovation on Firm Performance: Effects of Absorptive Capacity and Managerial Environmental Concern"

_sustainability, doi:10.3390/su11092455_

Round 1
Reviewer 1 Report
This is a very nicely written and innovative article, I must mention that especially bringing together absorptive capacity and green innovation is a very novel perspective that deserves more attention.
What I am missing in this article are the following points:
1) What is your methodology? You mention that you did a hierarchical linear modelling, but I don't see any methodology or any references, from which you adopted the methodology. I would strongly advise to have a brief, but compact section on methodology.
2) Limitations: Are there any limitations of the method chosen? You only discuss conceptual limitations, but not any methodological issues. Why is HLM the right choice of method in comparison to other methods such as mixed ANOVA, multiple regression or structural equation modelling? This is not clear and needs to be highlighted. Here, I would advise you to take a look at http://dx.doi.org/10.20982/tqmp.10.1.p013
3) Recent Findings: I admire the fact that you are using resources from Sustainability, the journal to which you submitted this work. I believe that the following paper can also be relevant for your research:
https://dx.doi.org/10.3390/su11030617
4) Theoretical Framework: Clearly, there is a link between WHY innovations have an effect on firm performance. Hitherto, the literature has focused on innovations as merely technological artefacts, as you also suggest - nevertheless, innovations without interactions to firm culture and performance, in other words, innovations without market shaping activities are nothing. I would suggest you to strengthen the emphasis on the link between these two concepts by making use of the following contributions:
Erkut, B. (2016): "Product Innovation and Market Shaping: Bridging the Gap with Cognitive Evolutionary Economics", IJM, 4(2); pp.3-24
https://doi.org/10.3390/joitmc4030023
5) Transfer value: You make a point by showing that the study merely focuses on China, which is one of the biggest economies of our time and an efficiency-driven one according to the GCI. I do not agree with you that there is no transfer value of the article because it merely focuses on China. On the contrary, we can make some thoughts by focusing on how structurally similar China is in terms of innovation to other economies. I want you to make some thoughts on structrural similarities of China with other efficiency-driven economies (DOI:10.18559/SOEP.2016.5.6) and what can be the implications for transfer value and/or for future research.
I wish you best of luck with their interesting and novel research direction!
Author Response
Dear Reviewer,
Please, find the attached document for your perusal.
Thank you for spending your valuable time on this manuscript.

Reviewer 2 Report
From my point of view the authors have carried out a rigorous and complete research project about an interesting and important issue. Also, the authors have made an important effort to explain this research project in this well written and well organized article. However, in my opinion, this article is not ready for being published yet because the following issues should be addressed before publication:
- From my point of view, in the first part of the article, in the introduction and section 2, the material is presented in a too much extended way, much more like a doctoral dissertation than a scientific article. In my opinion, this way of presenting it, will make difficult for potential readers to understand your article and find your research main novelty and contribution. I suggest you try to summarize and systematize this part by reducing its extension and adding tables and figures.
- In this sense the conclusions could also improve by being more direct and systematic and not having a subsection.
- Absorptive is defined in 2.3, in line 183, but appears for the first time in line 78
- I also suggest to add some appendixes with some material like the questionnaire, information about the companies such as their location within China, their sector/main activities…
- There is some editing needed, “ in line 425; also some strange spaces in lines 354, 446…
Author Response

(The authors gave the same response as above.)

Round 2
Reviewer 2 Report
From my point of view the authors have carried out a rigorous and complete research project about an interesting and important issue. Also, the authors have made an important effort to explain this research project in this well written and well organized article. This article has improved from the last version. However, in my opinion, this article is not ready for being published yet because the following issues should be addressed before publication:
- From my point of view, in the first part of the article, in the introduction and section 2, the material is presented in a too much extended way, much more like a doctoral dissertation than a scientific article. In my opinion, this way of presenting it, will make difficult for potential readers to understand your article and find your research main novelty and contribution. I suggest you try to summarize and systematize this part by reducing its extension and adding tables and figures. For example a table of references which is used successfully in most research papers.
- In this sense the conclusions should also improve by being more direct and systematic. From my point of view they are now even longer and they should be separated in two sections, for example discussion and conclusions like there was in the paper before. Papers with straight forward conclusions are easier to understand, readers find easier to know the main novelty and contribution of the research project to its field of knowledge.
- Absorptive is defined in 2.3, in line 183, but appears for the first time in line 78. This difficults understanding the article.
Author Response
Dear Reviewer,
Thank you for your useful suggestions. Please, find the attached document for your perusal.
Thanks for your co-operation.
